# The Legacy of the Idrija Mine Twenty-Five Years after Closing: Is Mercury in the Water Column of the Gulf of Trieste Still an Environmental Issue?

**DOI:** 10.3390/ijerph181910192

**Published:** 2021-09-28

**Authors:** Elena Pavoni, Elisa Petranich, Sergio Signore, Giorgio Fontolan, Stefano Covelli

**Affiliations:** 1Dipartimento di Matematica e Geoscienze, Università Degli Studi di Trieste, Via Weiss 2, 34128 Trieste, Italy; epavoni@units.it (E.P.); epetranich@units.it (E.P.); fontolan@units.it (G.F.); 2Autorità di Sistema Portuale del Mare Adriatico Orientale-Porto di Trieste, Via Karl Ludwig Von Bruck 3, 34144 Trieste, Italy; sergio.signore@porto.trieste.it

**Keywords:** mercury, sediments, water column, suspended particulate matter

## Abstract

Mercury (Hg) contamination in the Gulf of Trieste (northern Adriatic Sea) due to mining activity in Idrija (Slovenia) still represents an issue of environmental concern. The Isonzo/Soča River’s freshwater inputs have been identified as the main source of Hg into the Gulf, especially following periods of medium-high discharge. This research aims to evaluate the occurrence and distribution of dissolved (DHg) and particulate (PHg) Hg along the water column in the northernmost sector of the Gulf, a shallow and sheltered embayment suitable for the accumulation of fine sediments. Sediment and water samples were collected under unperturbed and perturbed environmental conditions induced by natural and anthropogenic factors. Mercury in the sediments (0.77–6.39 µg g^−1^) and its relationship to grain size were found to be consistent with previous research focused on the entire Gulf, testifying to the common origin of the sediment. Results showed a notable variability of DHg (<LOD–149 ng L^−1^) and PHg (0.39–12.5 ng L^−1^) depending on the interaction between riverine and marine hydrological conditions. Mercury was found to be mainly partitioned in the suspended particles, especially following periods of high discharge, thus confirming the crucial role of the river inputs in regulating PHg distribution in the Gulf.

## 1. Introduction

Among potentially toxic trace elements (PTEs) found in the environment, mercury (Hg) is a focus of global concern and was included among the World Health Organization’s top ten “chemicals of concern” in 2017 [1].

Mining activity and related mineral processing, as well as coal combustion and industrial activities (e.g., chlor-alkali plants), are generally considered among the major anthropogenic sources of Hg [2] and other PTEs in the environment. Atmospheric deposition, erosion and riverine inputs of suspended particulate matter (SPM) contribute to convey Hg in estuaries and marine-coastal areas where the element is accumulated in the bottom sediments [3,4]. Indeed, the sediment compartment may act both as a sink and a secondary source of contamination due to resuspension events and remobilisation processes with the subsequent release of both dissolved and particulate Hg species into the water column [5,6].

Moreover, the top few centimetres of sediment often represent the primary site for the production of methylmercury (MeHg) [7,8,9,10], the organic form of Hg of main concern due to its high toxicity and bioaccumulation potential in the aquatic food chain [8,11,12,13].

In the offshore marine sediments of the Mediterranean Sea, Hg reaches concentrations (avg. 0.10–0.20 µg g^−1^, [14]) which testify to an enrichment with respect to the world-wide natural background (0.03 µg·g^−1^) [15] due to both natural and anthropogenic sources. Among the most contaminated areas in Italy, the Venice Lagoon, the Marano and Grado Lagoon and the Gulf of Trieste are located in the northern area of the Adriatic Sea [16,17,18,19,20] where marine sediments showed the highest amount of Hg (0.05–8.63 µg·g^−1^, [21]), diminishing towards the central and southern sector of the same basin [21].

It has been demonstrated that Hg contamination in the Gulf of Trieste mostly originates from riverine inputs of the Isonzo/Soča River e.g., [19,22,23] draining Hg-enriched river banks and floodplain deposits along the river basin hosting the second largest Hg mine worldwide [24,25,26]. Indeed, over 500 years of *cinnabar* (HgS) extraction activity at the Idrija mining district (western Slovenia, Figure 1) has led to Hg contamination of multiple environmental matrices both in the areas surrounding the mine and far from this primary source [24,25,27,28,29].

In the Gulf of Trieste, the occurrence and behaviour of Hg have been the main topics of several studies focused on coastal sediment contamination [19,30], transport and distribution of Hg associated with the SPM at the Isonzo River mouth [22,23] as well as Hg cycling at the sediment-water interface [31,32]. However, little information is currently available on the occurrence of dissolved (DHg) and particulate (PHg)Hg in the northernmost sector of the Gulf of Trieste (Bay of Panzano), where the main access channel approaching the Port of Monfalcone is located. This embayment represents a suitable environment for the accumulation of suspended particles enriched in Hg.

Moreover, there is growing interest on the part of national and local authorities regarding the potential impact of Hg in the water column related to the resuspension of sediments due to future dredging operations needed to allow the navigation of ships approaching the port area.

In this context, the primary aim of this research is to evaluate the occurrence of Hg in the surface sediments as well as its partitioning behaviour between solid and dissolved phases along the water column under both unperturbed and perturbed environmental conditions, the latest induced by both natural and anthropogenic factors. This study provides a snapshot of the present situation and a baseline for Hg in the water column, useful for future evaluation of the impact of Hg in this coastal environment.

## 2. Materials and Methods

### 2.1. Environmental Setting

The Gulf of Trieste is a semi-closed shallow-water basin located in the northern Adriatic Sea with a maximum water depth of 25 m in its central sector. Water salinity in the Gulf typically ranges between 25 and 38 PSU (using the Practical Salinity Scale), whereas the seawater temperature ranges between 5 and 26 °C following the seasons [33]. The water circulation in the Gulf is mainly dominated by the anticlockwise circulation pattern of the Adriatic Sea and is controlled by tides, seasonal variations in the freshwater inflow, and winds (Bora E-NE, Libeccio SW and Scirocco SE) [34], which significantly influence the vertical water circulation [35].

In the Gulf of Trieste, the Isonzo/Soča River is the main input of both freshwater (average discharge of 83 m^3^·s^−1^, [33]) and SPM, whose distribution is regulated by the interaction between meteo-marine and riverine hydrological conditions.

The Isonzo River is known as the primary source of Hg into the Gulf of Trieste and the element at the river mouth was found to be almost completely partitioned in the SPM [36]. In this context, the river freshwater inputs play a crucial role in the occurrence and distribution of PHg both under periods of medium-high river discharge [22] and during extreme river plume events when the influx of PHg into the Gulf ranged between 37.0 and 112 ng·L^−1^ [23].

Regarding the surface sediment of the Gulf of Trieste, the highest concentrations of Hg were found at the Isonzo River mouth (23.3 µg·g^−1^) due to the prevalence of *cinnabar* particles in the coarser sandy-silty fraction of the sediment [19,37]. Although the amount of Hg in the sediment decreases with increasing distance from the river mouth [19], extensive Hg contamination may also be present in the nearshore areas of the northernmost sector of the Gulf of Trieste (Bay of Panzano), a shallow and sheltered embayment promoting the accumulation of fine sediments and contaminants (Figure 1). The area is affected by several anthropogenic activities including agricultural and industrial settlements in the hinterland and tourist and mussel farming areas along the coast. An additional source of potential contamination is represented by the city of Monfalcone, which is home to a thermoelectric plant, several coal, petroleum and other cargo handling equipment and an extended port area which can be reached through a main channel located between the Isonzo River mouth and a mussel farm (Figure 1). Industrial activity in the port area was thought to be a potential source of organic contaminants (PAH and PCB) in the sediments of the Bay of Panzano [38] and residues from antifouling paints used on boats have been identified as a source of PTEs (e.g., Cu and Zn) [38,39].

### 2.2. Sampling Strategy

Sampling operations for the collection of sediment and water samples were performed at six sites (P1–P6) located in the vicinity of the main access channel to the Port of Monfalcone (Figure 1). With the exception of site P6 (located in the offshore marine area of the Bay of Panzano), all the sampling stations are representative of different targets: mussel farming (P1, P2 and P3), marine phanerogam meadows (P4) and tourist attractions along the beach (P5). Moreover, site P4 is located in the marine coastal sector adjacent to a confined disposal site for the storage of dredged sediments (Figure 1). These targets could be affected by both the Isonzo River plume events and the resuspension of fine Hg enriched particles induced by natural and anthropogenic factors (e.g., dredging).

Daily average discharge from the Isonzo River at the time of sampling was recorded from the gauging station of Pieris (Gorizia) located approximately 15 km upstream from the river mouth (Table 1). Vertical profiles of salinity (PSU), temperature (°C) and turbidity (NTU) were recorded by means of a CTD multiprobe (Hydrolab H20 Multiprobe, OTT HydroMet, Loveland, CO, USA) with a 0.10 dbar pressure step and a sampling rate of 1 s) before sampling. Two water samples were collected using a Niskin bottle (Hydro-Bios Apparatebau GmbH, Altenholz, Germany) from the surface (0–0.5 m depth) and bottom (0.50 m from the bottom sediment) water layers, respectively. Sampling operations were performed during five sampling campaigns carried out under different environmental conditions including (i) unperturbed conditions characterised by low river flow, the absence of wind and good weather (sampling campaigns 1 and 4); (ii) perturbed conditions induced by natural factors such as periods of moderate-high river discharge (sampling campaigns 2 and 5) and conditions of windy sea (sampling campaign 3); (iii) perturbed conditions induced by anthropogenic activities (the movement of ships, sampling campaign 6) (Table 1). Although sampling campaigns 2 and 5 were both performed following a period of moderate-high river discharge, it should be pointed out that the river discharge was notably low during the sampling campaign 5 (87.3 m^3^ s^−1^) compared to sampling campaign 2 (328 m^3^ s^−1^), which was performed following a period of particularly heavy river flow (Figure 1; Table 1).

Water samples for the analytical determination of DHg were filtered (Millipore Millex HA, 0.45 µm pore size, Millipore, Burlington, MA, USA) in the field, collected into pre-conditioned borosilicate glass containers and immediately oxidised by adding bromine chloride (BrCl, Hg-free from Brooks Rand Instruments, Seattle, WA, USA, 0.5 % *v/v*, until the sample turned the colour yellow) according to the EPA Method 1631e [40]. Additional 2 L water samples were taken to the laboratory where the SPM was separated from the dissolved fraction by vacuum filtration.

During sampling campaign 1, surface sediments were also collected at each site (P1–P6) using a stainless steel Van Veen grab (1.7 L, Hydro-Bios Apparatebau GmbH, Altenholz, Germany)). Three distinct aliquots of sediment were collected and a stainless steel spoon was employed to rapidly scrape off the first 2 cm of the sediment surface which was then homogenised in situ to get a composite sample, stored in appropriate containers and transported to the laboratory.

In addition, three multiprobes were placed at sites P2 (approximately 2 m depth and at the bottom, Aanderaa RCM9, Aanderaa Data Instruments AS, Bergen, Norway) and P3 (approximately 2 m depth, Hydrolab DS5 OTT HydroMet, Loveland, CO, USA) in order to achieve in situ continuous measurements of temperature (°C), salinity (PSU, Practical Salinity Unit) and turbidity (NTU, Nephelometric Turbidity Unit) along the water column.

Sampling campaign 6 was performed at different sites (A-E and P3) towards the main access channel to the Port of Monfalcone in order to compare unperturbed and perturbed conditions which occurred before and after a large draught ship (8 m) had entered and subsequently left the area (Figure 1; Table 1). To achieve this objective, the area (site A) located between the mussel farm (site P3) and the navigation channel (site B) was selected as the most representative (Figure 1). There (sites A, B and P3), as well as along the main channel to the port area (sites C–E), turbidity vertical profiles were recorded before and after a ship had entered and left the area.

In detail, the unperturbed condition was evaluated by means of turbidity vertical profiles recorded at sites P3 and B before the ship had entered the selected area. After the ship had passed by, turbidity profiles were recorded approximately every 10 min at site A in order to evaluate variations in the turbidity values along the water column over time. In addition, two water samples for the analytical determination of DHg and PHg were collected at site A, one at the bottom and one at approximately 7 m depth, where the maximum turbidity zone was observed. Subsequently, turbidity vertical profiles were also recorded following the ship at sites C, D and E, towards the main channel to the port area and once again at site P3 where additional water samples were collected at the bottom and at approximately 7 m depth, respectively.

### 2.3. Surface Sediments: Grain Size Analysis and Total Hg Content

For grain size analysis, 15–20 g of fresh sediment sample were processed using hydrogen peroxide (H_2_O_2_, 10%) for 24 h to eliminate most of the organic matter, and then wet-sieved using a 2 mm sieve. The resulting < 2 mm fraction was analysed by means of a laser granulometer (Malvern Mastersizer 2000, Malvern Panalytical Ltd., Malvern, UK).

A subsample of the sediment was frozen and freeze-dried (CoolSafe 55-4 SCANVAC, Scientific Laboratory Supplies Ltd., Nottingham, UK), homogenised and ground for Hg determination. Total Hg was determined by means of a Direct Mercury Analyser (DMA-80, Milestone, Sorisole, Italy) according to the EPA Method 7473 [41]. Three replicates were analysed for each sediment sample and the quality of the analysis was evaluated by means of certified reference material (PACS-3 Marine Sediment CRM, NRCC, Whitehorse, YT, Canada), obtaining acceptable recoveries ranging between 88 and 101%. The limit of detection (LOD) was approximately 0.005 ng and the precision of the analysis expressed as RSD% was <2%.

### 2.4. Analytical Determination of Particulate and Dissolved Hg

The SPM concentrations were determined by vacuum filtration on pre-conditioned and pre-weighed (Mettler, precision 0.00001 g) Millipore HA membrane filters (ø 47 mm, 0.45 µm pore size). Filters were dried at room temperature to avoid Hg^0^ volatilisation due to heat sources and then stored in air-tight containers over silica gel for 4–5 days, thereby protecting them from humidity in the air. Filters were acid-digested in a closed microwave system (Multiwave PRO, Anton Paar GmbH, Graz, Austria) using *aqua regia* (suprapure HCl ≥ 37% VWR and HNO_3_ ≥ 69% VWR, 3:1) following the modified EPA Method 3052 [42]. The obtained solutions were diluted up to a volume of 25 mL by adding Milli-Q water and appropriately stored before analysis.

The analytical determination of Hg in the dissolved (DHg) and in the SPM (PHg) fractions was performed by means of Cold Vapor Atomic Fluorescence Spectrometry coupled with a gold trap preconcentration system (CV-AFS Mercur, Analytic Jena GmbH, Jena, Germany). Water samples were analysed following the EPA Method 1631e [40] which requires a pre-reduction using NH_2_OH-HCl (250 µL/100 mL sample) until the yellow colour disappeared, followed by a reduction with SnCl_2_ (Sigma-Aldrich 2% in HCl 2%). The instrument was calibrated using standard solutions obtained via dilution from NIST 3133 certified solution and acidified with BrCl (0.5%, *v/v*). Certified reference material (ORMS-5 CRM, Brantford, ON, Canada) was analysed in the same batch as the water samples for quality control and an acceptable recovery was obtained (105%). The limit of detection was 0.60 ng L^−1^ and the precision of the analysis expressed as RSD% was < 3%.

### 2.5. Exploratory Multivariate Data Analysis

Principal component analysis (PCA) was used as an unsupervised exploratory chemometric tool to evaluate the relationships within samples (PC scores and score plot), within variables (PC loadings and loading plot) and between samples and variables (biplot) [43]. In detail, PCA was performed on physico-chemical parameters (salinity, temperature, SPM concentration and river discharge), PHg and DHg observed at the six investigated sites (P1–P6) under different environmental conditions (sampling campaigns 1–5). Column autoscaling was applied to data matrices to minimise systematic differences between variables [44] and multivariate data processing was performed using the CAT (Chemometric Agile Tool) package, based on the R platform (The R Foundation for Statistical Computing, Vienna, Austria) and freely distributed by Gruppo Italiano di Chemiometria (Italy) [45].

## 3. Results and Discussion

### 3.1. Physico-Chemical Parameters of the Water Column

Riverine inputs of suspended particles play a major role in the transport of Hg and other PTEs in estuarine and marine-coastal environments [46,47,48]. In these ecosystems, the composition of the SPM may be affected by several factors including hydrodynamic conditions, interactions between freshwater and saltwater, adsorption/desorption processes, sedimentation and resuspension of bottom sediments [46]. In this context, the physico-chemical boundary conditions along the water column (e.g., temperature, salinity, turbidity, pH, redox potential, dissolved oxygen) may affect Hg partitioning behaviour between solid and dissolved phases as well as its speciation, mobility and bioavailability [47].

A summary of the basic physico-chemical parameters (salinity, temperature and turbidity) measured along the water column at the six investigated sites (P1–P6) under different environmental conditions (sampling campaigns 1–5) is reported in Appendix A. Two distinct water masses were observed under unperturbed conditions (sampling campaigns 1 and 4) as a result of the interaction between river freshwater and seawater. Although slightly higher salinity values were recorded in the surface water in April (sampling campaign 4) at sites P2, P3 and P4 (31–33 PSU), brackish salinity values were generally observed at the other sites (22–28 PSU) increasing with depth and reaching typical marine salinity values at the bottom (36–37 PSU).

The river freshwater input was especially evident in March during sampling campaign 3 at site P6 and sampling campaign 2, which was performed following a period of intense discharge from the Isonzo River. Indeed, brackish water down to a depth of 1 m (ranging overall between 14 at site P1 and 26 at site P6) along with a sharp deeper halocline was observed at all the investigated sites (Appendix A).

Conversely, brackish water (18 PSU) was observed only at sites P1 and P4 in May during sampling campaign 5, most likely due to a generally lower river discharge (87.3 m^3^ s^−1^ at the time of sampling) if compared to that seen in March (328 m^3^ s^−1^, sampling campaign 2) (Table 1; Figure 1).

Temperature showed slight variations along the water column and among different sampling campaigns (Appendix A). The lowest values were recorded in the surface water in February and March (10.9 ± 0.9 and 10.6 ± 0.5 °C during sampling campaigns 1 and 2, respectively) and comparable values were measured at the bottom (9.43 ± 0.23 and 10.2 ± 0.1 °C during sampling campaigns 1 and 2, respectively). Conversely, higher values of temperature were observed in April and May, both in the surface water (15.8 ± 0.5 and 15.8 ± 0.4 °C during sampling campaigns 4 and 5, respectively) and at the bottom (15.0 ± 0.5 and 14.6 ± 0.3 °C during sampling campaigns 4 and 5, respectively).

Turbidity showed relatively low values in February (sampling campaign 1) ranging between 1.10 and 21.0 NTU in the surface water (at sites P1 and P6, respectively) and generally decreased with increasing depth reaching values <10 NTU most likely due to mixing and dilution processes between different water masses. Surprisingly, relatively low values of turbidity were also observed during the sampling campaigns performed following periods of high and moderate discharge from the Isonzo River (Appendix A) with the only exception being site P2 in May (38.9 and 20.0 NTU in the surface and bottom water, respectively). The maximum turbidity values were observed in April (sampling campaign 4) in the surface water at sites P1 (67.1 NTU) and P2 (58.2 NTU), decreasing with increasing depth at each sampling site (Appendix A). In this case, the relatively elevated turbidity values may be related to enhanced biological activity during late spring [49], in particular at the mussel farm. The only exception was the vertical profile recorded at sites P3 and P5 where almost constant values of turbidity were observed along the water column (approximately 15 and 25 NTU).

Moreover, turbidity vertical profiles recorded before and after a large draught ship had passed by (sampling campaign 6, Figure 1 and Figure 2) showed that before the ship had approached, turbidity was found to be extremely low (<5 NTU) at sites P3 and B, testifying to unperturbed conditions. A clear increment of the turbidity values was evident immediately after the ship had passed site A and the maximum values (20–25 NTU) were recorded at approximately 7 m depth about 30 min after the ship had sailed out of the area (Figure 2). However, the perturbation induced by the movement of the ship did not reach particularly high values of turbidity and lasted only a brief period of time. Indeed, unperturbed conditions were restored in less than two hours as highlighted by the vertical profile recorded at site P3 (<5 NTU) (Figure 2). In this context, the characteristics of both the ship (e.g., draught, speed) and the location (e.g., water depth, distance to shore, sediment grain size) may represent the two main factors governing the amount of the resuspended material [50].

Additional information was provided by the measurements of the Isonzo River discharge as well as the continuous measurements of salinity, temperature and turbidity recorded at the beginning of March at sites P2 (surface and bottom) and P3 (surface) (Appendix A). The effects induced by the high river discharge at the beginning of March (471 and 391 m^3^ s^−1^) were clearly evident at both sites P2 and P3, where a decrease in the salinity values corresponded to a decrease in temperature in the surface water, most likely due to notable freshwater input. Regarding turbidity, relatively low values were observed in the surface water reaching maximum values of 9.60 (at site P2) and 15.6 NTU (at site P3) which appeared to persist for a brief period of time. Indeed, unperturbed conditions were rapidly restored according to the results obtained from the comparison between unperturbed and perturbed conditions before the ship had approached and after it had sailed out of the area (Figure 2 and Appendix A).

Conversely, a notable increase in the turbidity values was observed at the bottom at site P2 (maximum value of 112 NTU). However, this perturbation lasted approximately 24 h, suggesting that it may have been related to technical operations at the mussel farm such as the lowering of a boat’s anchor.

### 3.2. Surface Sediments: Grain-Size and Hg Content

The surface sediments were found to be heterogeneous in terms of grain-size composition, although those collected at the mussel farm (P1, P2 and P3) showed a very similar grain-size spectra and composition (Figure 3). According to the classification proposed by Shepard [51], the surface sediments consisted predominantly of silt (23.3–82.8%), followed by sand (5.01–73.8%) and clay (2.87–14.5 %). The silty fraction clearly prevailed in the sediment collected at the mussel farm (sites P1, P2 and P3), followed by the offshore marine sector (site P6) and, to a lesser extent, site P4. Conversely, the surface sediment collected at site P5 showed the highest content of sand (73.8%) most likely due to its location close to the coast and the relatively shallow waters (3–4 m) and high wave energy which favour the settling of coarser particles in suspension (Figure 3).

The Hg concentration in the investigated surface sediments varied between 0.77 (site P1) and 6.39 (site P6) µg g^−1^ and the grain-size composition was consistent with previous research focused on the main channel approaching the Port of Monfalcone [30] (Table 2) showing that the surface sediments were dominated by silt, and Hg ranged between 0.30 and 13.5 µg g^−1^, decreasing from the offshore area to the innermost sector of the access channel to the port area [30].

The concentration of Hg in the surface sediments investigated in this study (0.77–6.39 µg g^−1^, Figure 3) exceeded the Italian regulatory threshold limit of 0.30 µg g^−1^ (Decrees of the Italian Ministry of the Environment 260/2010 and 172/2015 according to EU Directive 2000/60/EC). Although the results from this study testified to a total Hg concentration in the surface sediments which remains of concern, speciation analyses performed on sediments collected along the main access channel to the Port of Monfalcone recently demonstrated that the element appeared to be strongly associated with the less mobile chemical fractions [30]. This suggested that most of the Hg in the investigated sediments was not available for MeHg production unless under conditions of anoxia [32]. Indeed, the methylation rate does not only depend on the total amount of Hg [17,19,30] since several factors (e.g., temperature, pH, Eh, dissolved oxygen) may also have a role in MeHg production [47,64].

Mercury values of the same order of magnitude were also reported for the surface sediments of the Bay of Panzano (1.40–5.54 µg g^−1^, [52]) as well as for the northern Adriatic Sea [21] (Table 2). Conversely, notably lower values were found both in the central and southern sector of the Adriatic Sea [21] as well as the northern Tyrrhenian Sea [58] and at other marine coastal areas and estuarine environments worldwide (Table 2).

The amount of Hg in the investigated surface sediments was comparable to that observed in the offshore sector of the Gulf (ranging between 0.10 and 11.7 µg g^−1^, [19]) and significantly lower with respect to the Isonzo River mouth where the highest concentrations of Hg were observed in previous research (ranging between 4.45 and 23.3 µg g^−1^, [19]) (Figure 3), and primarily related to the occurrence of the detrital form of Hg (*cinnabar* particles) [37].

According to the linear function displaying the relationship between the concentration of Hg and the percentage of the 2–16 µm grain size fraction proposed by previous research, two groups of samples were identified [19] (Figure 3). The first included sediments collected at the Isonzo River mouth, whereas the second referred to sediments from the whole Gulf. The surface sediments investigated in this study belonged to the second group, confirming their common origin with respect to the offshore sediments of the Gulf of Trieste (Figure 3).

### 3.3. Suspended Particulate Matter: Distribution and Hg Concentration

No notable differences in the SPM were observed at the six investigated sites and the highest values were observed during sampling campaign 2 which was performed following a period of generally high discharge from the Isonzo River (Appendix A). The surface-bottom SPM ratios were generally low and <1 both under unperturbed (0.87 ± 0.07, 0.63 ± 0.17 and 0.72 ± 0.17 during sampling campaigns 1, 3 and 4, respectively) and perturbed environmental conditions (sampling campaign 5, 0.78 ± 0.10). Conversely, high surface-bottom SPM ratios were observed following a period of high river discharge at the beginning of March (sampling campaign 2, 1.94 ± 0.94 with maximum values of 3.59 and 2.45 at sites P1 and P2, respectively) as a result of high freshwater and SPM inputs from the Isonzo River. This confirms that the SPM distribution in the investigated area depends heavily on the river discharge, as also suggested by the PCA output (Figure 4) and the significant correlation (N = 30, r = 0.734, *p* < 0.01; on average N = 5, r = 0.995, *p* < 0.01) observed between the average SPM concentration in the surface water at the six investigated sites and the Isonzo River discharge during the 5 sampling campaigns (Figure 5A).

The highest concentrations of PHg were observed under perturbed conditions during sampling campaign 2, both in the surface water (8.37 ± 2.11 ng L^−1^) and at the bottom (6.26 ± 1.62 ng L^−1^), especially at site P3 (12.5 and 8.64 ng L^−1^ in the surface water and at the bottom, respectively) (Figure 4 and Figure 6). Moreover, a moderate correlation was observed between PHg and the Isonzo River discharge (Figure 5B; N = 30, r = 0.644, *p* < 0.01; on average N = 5, r = 0.761, *p* < 0.5) as well as between PHg and the SPM concentration (Figure 5C; N = 30, r = 0.634, *p* < 0.01; on average N = 5, r = 0.927, *p* < 0.1) confirming the role of the Isonzo River as the primary source of Hg which enters the Gulf, mainly in the form of SPM, as highlighted by previous research [22,36,65]. Indeed, it has been demonstrated that the dispersion of Hg from the Isonzo River mouth depends heavily on the interaction between riverine and meteo-marine hydrological conditions and occurred following four principal directions, including in the direction of the Port of Monfalcone [19]. Consequently, suspended particles enriched in Hg were trapped in the Bay of Panzano, especially when winds such as the Scirocco and Libeccio are dominant.

In this study, the maximum river discharge (925 m^3^ s^−1^) was reached approximately three weeks before sampling, remaining relatively elevated (131–471 m^3^ s^−1^) for several days after sampling campaign 2. Accordingly, sampling campaign 5 was performed when the river discharge was moderate (87.3 m^3^ s^−1^) and slightly lower values of PHg were observed (1.11 ± 0.66 and 2.42 ± 1.15 ng L^−1^ in the surface and bottom water layers, respectively) (Figure 6). Moreover, the amount of PHg observed during sampling campaign 5 was found to be comparable to that of the sampling campaigns performed both under conditions of windy sea (sampling campaign 3) and under unperturbed conditions (sampling campaigns 1 and 4) (Figure 4 and Figure 6). This suggests that relatively elevated values of PHg in the investigated area may be restricted to brief periods of particularly intense discharge from the Isonzo River, as evidenced by the PCA output (Figure 4) and the correlation between PHg and the river discharge (Figure 5B). Indeed, river flooding is responsible for large inputs of freshwater, SPM and particulate-associated contaminants [66,67]. In this context, it has been demonstrated that notable concentrations of PHg were discharged into the Gulf of Trieste during extreme Isonzo River flood events (maximum value of 49 µg g^−1^ following a river discharge of 1600 m^3^ s^−1^, [65]).

Most likely due to dilution effects between riverine freshwater and saltwater, the PHg concentrations at the six investigated sites were generally found to be notably low with respect to those observed at the Isonzo River mouth by previous research [22,36], in front of the river mouth within a buoyant river plume [23] as well as along the Aussa River, flowing in the adjacent Marano Lagoon, which was affected by the discharge of Hg from a chlor-alkali plant [17,55,68] (Table 3). Conversely, PHg values were found to be of the same order of magnitude with respect to the offshore marine area of the Gulf during a river plume event [23].

As previously mentioned, a ship with a large draught (8 m) moving through the main access channel to the port area may temporarily affect the turbidity vertical distribution along the water column, reaching the maximum values at approximately 7 m depth at site A (Figure 2). The water sample collected at the maximum turbidity zone showed a PHg concentration of 14.0 ng L^−1^, which was two orders of magnitude higher than the PHg at the bottom (0.55 ng L^−1^) (Figure 6). Approximately 2 h after the ship had sailed out of the area, the PHg concentration at the same depth (approximately 7 m) at site P3 was notably lower (2.01 ng L^−1^), confirming that unperturbed conditions were restored after a brief period of time. However, a higher PHg concentration was observed at the bottom (11.8 ng L^−1^), most likely due to the settling of fine Hg enriched particles (Figure 6).

### 3.4. Dissolved Hg

The occurrence of DHg at the six sampling sites was investigated during unperturbed and perturbed environmental conditions both in the surface water and at the bottom (Figure 7; Appendix A). The highest concentrations of DHg were detected in winter both under unperturbed (sampling campaign 1, 25.9 ± 10.2 and 35.9 ± 37.4 ng L^−1^ in the surface water and at the bottom, respectively; Figure 4) and perturbed conditions (sampling campaign 2, 16.5 ± 16.4 and 40.1 ± 59.5 ng L^−1^ in the surface water and at the bottom, respectively), reaching the maximum concentration in the bottom saltwater at sites P5 (sampling campaign 1, 112 ng L^−1^) and P2 (sampling campaign 2, 149 ng L^−1^) (Figure 4 and Figure 7). Dissolved Hg concentrations were one order of magnitude lower in spring both under unperturbed (sampling campaign 4, 4.74 ± 1.59 and 5.20 ± 7.31 ng L^−1^ in the surface water and at the bottom, respectively) and perturbed conditions (sampling campaign 5, 4.21 ± 3.17 and 3.72 ± 1.86 ng L^−1^ in the surface water and at the bottom, respectively) (Figure 7).

Conversely, DHg was mainly <LOD during the sampling campaign performed at the end of March (sampling campaign 3) (Figure 7), most likely due to the intense windy conditions during sampling operations. Indeed, turbulence induced by wind [74] and the subsequent mixing of the water column [75] may promote the release of gaseous elemental Hg from the surface water to the atmosphere, although the highest Hg evasion was found to occur in summer [48,76]. Moreover, sampling campaign 3 was performed following a period of low discharge from the Isonzo River (<100 m^3^ s^−1^) and relatively low PHg concentrations were observed (1.42 ± 0.97 and 2.81 ± 2.51 ng L^−1^ in the surface and bottom water layers, respectively). This suggests that low amounts of PHg were available to desorption and or dissolution processes with subsequent limiting of Hg release from the suspended particles to the dissolved fraction.

Dissolved Hg concentrations were of the same order of magnitude with respect to previous research focused on the Isonzo River mouth [22] and notably higher than those reported for the Gulf of Trieste [69] (Table 3). Generally, DHg at the investigated sites was higher compared to other aquatic systems along the Portuguese coast [72], the Tagus estuary [71], Tinto and Odiel estuaries [61] as well as the Gulf of Cádiz in Spain [61], especially in the bottom saltwater (Table 3).

As in the case of PHg, DHg reached a concentration of 19.0 ng L^−1^ at approximately 7 m depth where the maximum turbidity zone was observed at site A after a ship had left the area during sampling campaign 6, decreasing at the bottom (9.40 ng L^−1^). Notably lower values were found both at 7 m depth (7.11 ng L^−1^) and at the bottom (2.35 ng L^−1^) at site P3 approximately 2 h after a ship had passed by (Figure 7).

### 3.5. Mercury Partitioning between the Suspended Particulate Matter and the Dissolved Fraction: Distribution Coefficients (K_D_)

In aquatic systems, the partitioning behaviour of trace elements is mainly governed by adsorption/precipitation and desorption/dissolution processes. Indeed, trace elements can be preferentially associated with suspended particles (solid phase) and the dissolved fraction [77,78,79]. In this context, distribution coefficients (*K_D_*, L kg^−1^) are commonly employed to investigate trace element partitioning behaviour, although information regarding the element chemical form is not provided by this index.

According to the Equation (1), Hg distribution coefficients (*K_D_*, L kg^−1^) were calculated as the ratio between Hg concentration in the SPM (*PHg*, µg g^−1^) and in the dissolved fraction (*DHg*, ng L^−1^) and expressed on a logarithmic scale [77] (Table 4):(1)$$LogK_{D}=\left[PHg\right]/\left[DHg\right]$$

At the six investigated sites and under different environmental conditions, logK_D_ values ranged overall between 3.92 and 6.42 and between 3.69 and 6.68 in the surface and bottom water layers, respectively (Table 4).

The logK_D_ values were relatively high, thus testifying to the preferential partitioning of Hg in the suspended particles as also observed at the Isonzo River mouth [36] and other similar environments such as the New York/New Jersey Harbor, where Hg was found to be mainly associated with the SPM (98–99%) [7]. In addition, the distribution of DHg in the surface water did not appear to be simply governed by salinity, since Hg is generally high particle reactive and easily involved in removal processes through adsorption and/or precipitation [7,11,48,61]. A significant correlation was observed between logK_D_ and DHg (N = 60, r = 0.897, *p* < 0.01), especially following periods of high freshwater discharge from the Isonzo River (sampling campaigns 2 and 5) (Figure 8), although the logK_D_ values did not notably vary among different sampling conditions.

## 4. Conclusions

The occurrence of Hg in the coastal area of the Gulf of Trieste still remains an issue of environmental concern, although extraction activities at the Idrija Hg mine (Slovenia) ceased in 1996.

Results from this research confirmed the role of the Isonzo River as the primary source of both dissolved and particulate Hg in the northernmost sector of the Gulf of Trieste (Bay of Panzano), especially following periods of high discharge of the river. However, contrary to DHg, which showed both a notable spatial and temporal variability (<LOD–149 ng L^−1^), the amount of PHg (0.39–12.5 ng L^−1^) appeared to be strongly related to the river inputs of freshwater and SPM. Indeed, the highest amounts of PHg both in the surface water and at the bottom were found to be restricted to brief periods of intense river discharge. In agreement with previous investigations, Hg in the water column was still found to be mainly partitioned in the SPM, as also confirmed by the elevated logK_D_ values (3.69–6.68), thus testifying to its behaviour as showing a high affinity for fine sediment particles transported in suspension.

At the investigated area in the Bay of Panzano, the relatively shallow water depth allowed PHg accumulation in the surface sediments, which showed remarkable Hg concentrations (0.77–6.39 µg g^−1^). However, the amount of Hg in the sediments was found to be notably low with respect to the littoral zone surrounding the Isonzo River mouth and of the same order of magnitude if compared to sediments from the offshore sector of the Gulf of Trieste.

Resuspension events caused by natural and anthropogenic factors certainly affect the mobility of Hg from the sediment compartment to the water column, but results from this research showed that they can be limited. Indeed, perturbed conditions along the water column due to the presence of a large draught ship approaching the port area lasted only a brief period of time (approximately 2 h). The observed increase for both PHg and DHg in the water column is temporary since unperturbed conditions were promptly restored. This evidence suggests that a similar scenario would also occur for dredging activities where the effect in terms of widespread Hg in the water column should be restricted both to the operation area and time period.

Results from this research also suggested that the magnitude of a natural event, such as the increase in wave motion or extreme Isonzo River flood events, would alter DHg and PHg concentrations in the water column more significantly than a local perturbation caused by anthropogenic activities. Moreover, considering the degree of contamination reported for the Isonzo River basin [80,81], it may be expected that the metal will continue to be transported from inland to the Gulf’s waters for the foreseeable future.

Since the Isonzo River’s discharge conditions were identified as a crucial factor in regulating the amount of Hg in the northernmost sector of Gulf of Trieste, future research should address the effects of variations in the Isonzo River discharge on the contribution of Hg associated with the SPM as well as the evaluation of Hg and SPM fluxes in the investigated area.

## Figures and Tables

**Figure 1 ijerph-18-10192-f001:**
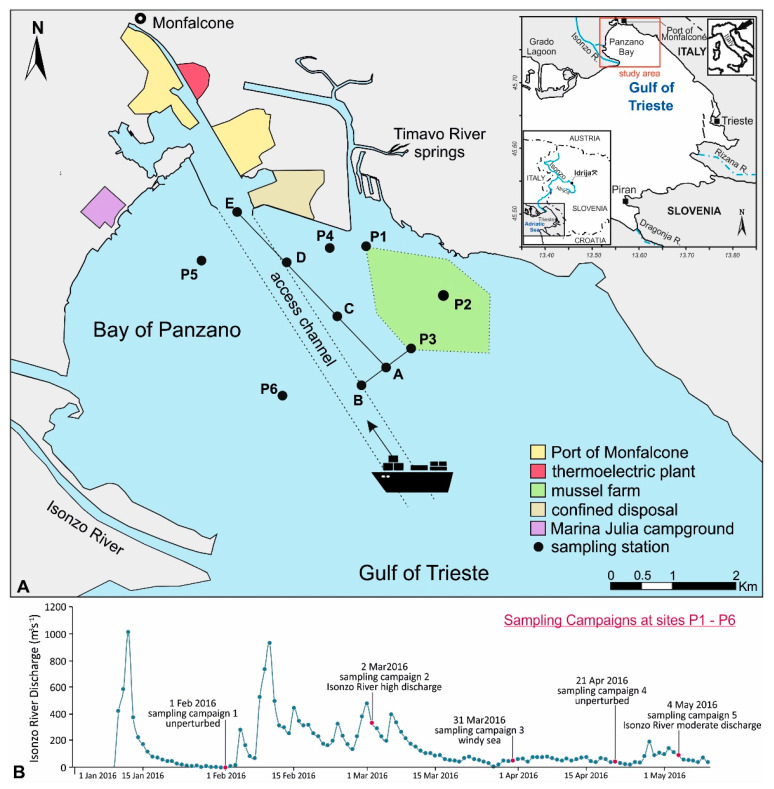
(**A**). Map of the study area and location of the sampling sites near the main access channel to the Port of Monfalcone (Bay of Panzano, Gulf of Trieste). (**B**). Isonzo River daily discharge (m^3^ s^−1^) from 1 January 2016 to 10 May 2016.

**Figure 2 ijerph-18-10192-f002:**
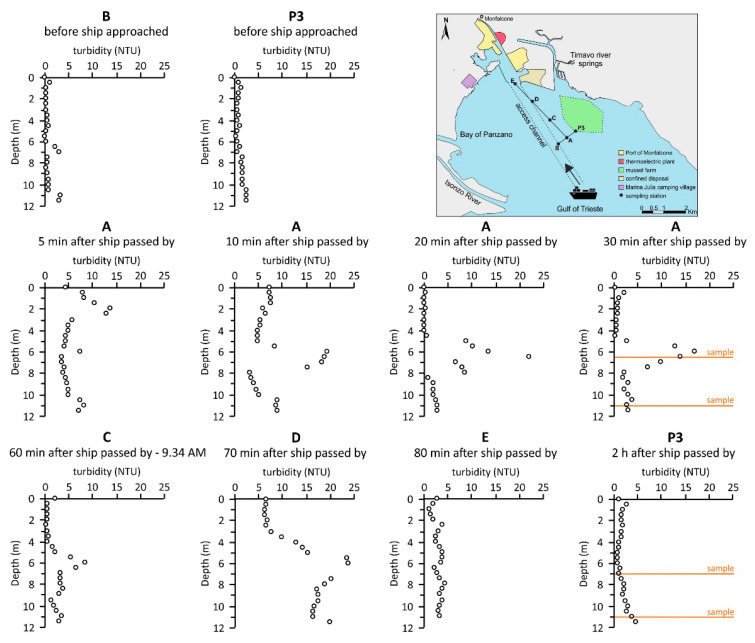
Turbidity (NTU) vertical profiles recorded at different sites in the area located between site P3 and the navigation channel before a large draught ship had entered and after it have left the area towards the main access channel to the Port of Monfalcone (sampling campaign 6). Water samples were collected at different depths (yellow lines) at sites A and P3 for the determination of both DHg and PHg.

**Figure 3 ijerph-18-10192-f003:**
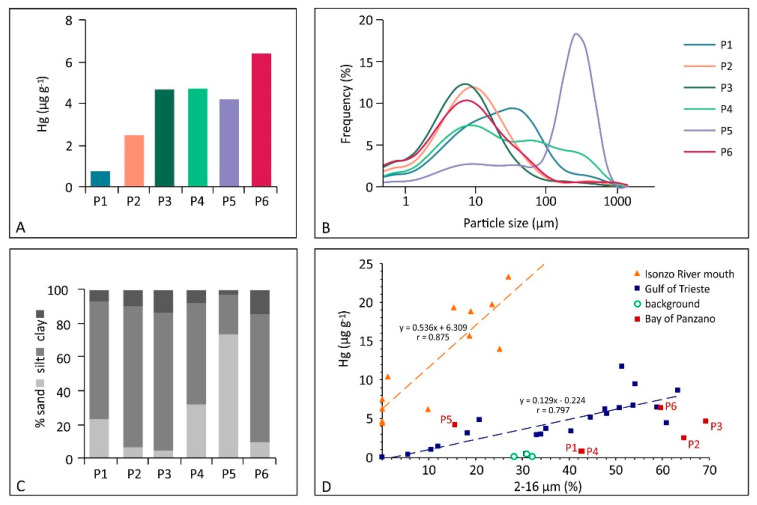
(**A**) Concentration of Hg (µg g^−1^) in the surface sediments collected at the six investigated sites (P1–P6) in the Bay of Panzano; (**B**) Grain size spectra of the sediment samples; (**C**) Grain size composition of the sediment samples; (**D**) Relationship between Hg concentration in the surface sediments and the 2–16 µm grain size fraction (modified and redrawn from [19]).

**Figure 4 ijerph-18-10192-f004:**
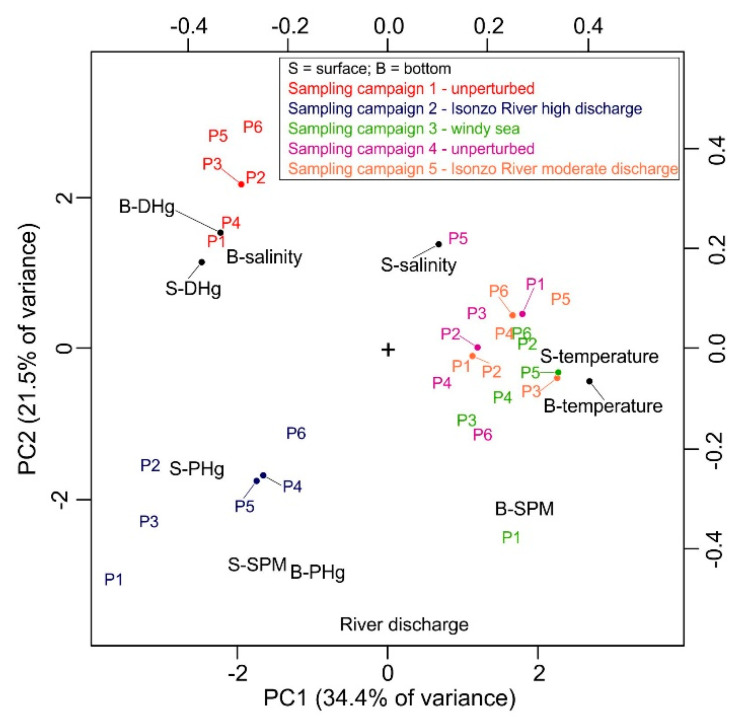
Biplot summarising the results of PCA performed on physico-chemical parameters (salinity, PSU; temperature, °C; concentration of SPM, mg L^−1^) dissolved and particulate Hg (DHg and PHg, ng L^−1^) observed in surface and bottom water samples collected at the six investigated sites (P1–P6) in the Bay of Panzano during sampling campaigns 1–5 performed under different environmental conditions.

**Figure 5 ijerph-18-10192-f005:**
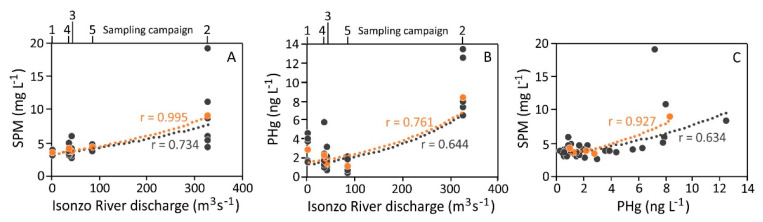
Correlation between the Isonzo River discharge (m^3^ s^−1^) and (**A**) SPM concentration (mg L^−1^), (**B**) PHg concentration (ng L^−1^) and (**C**) between the SPM concentration (mg L^−1^) and PHg concentration (ng L^−1^) in the surface water layer at the six investigated sites in the Bay of Panzano. Average SPM and PHg values and correlations are plotted in orange. Best fit was obtained using a 2nd-order polynomial curve.

**Figure 6 ijerph-18-10192-f006:**
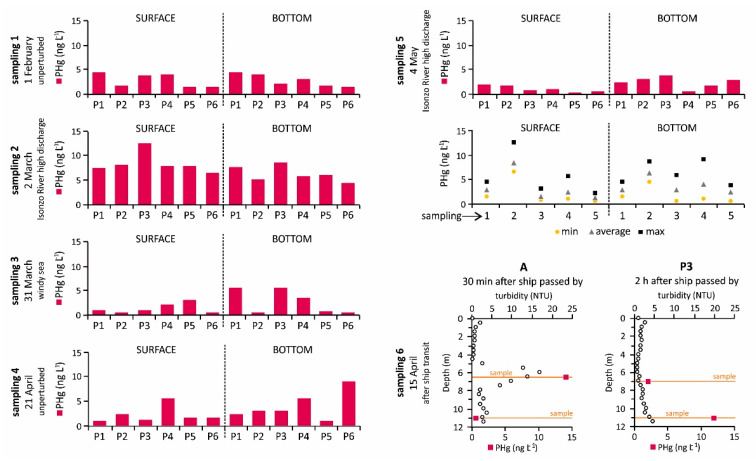
Concentration of PHg (ng L^−1^) in the surface and bottom water samples collected at the six investigated sites in the Bay of Panzano under different environmental conditions during sampling campaigns 1–5 and concentrations of turbidity (NTU) and PHg (ng L^−1^) along the water column at sites A and P3 after a ship had sailed out of the area during sampling campaign 6.

**Figure 7 ijerph-18-10192-f007:**
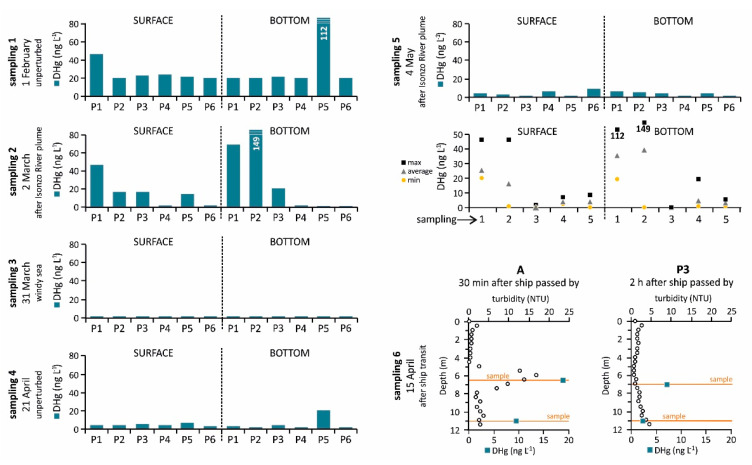
Concentration of DHg (ng L^−1^) in the surface and bottom water samples collected at the six investigated sites in the Bay of Panzano under different environmental conditions. The “°” symbol represents the single measurements of turbidity, as reported in the *x*-axis in the figure.

**Figure 8 ijerph-18-10192-f008:**
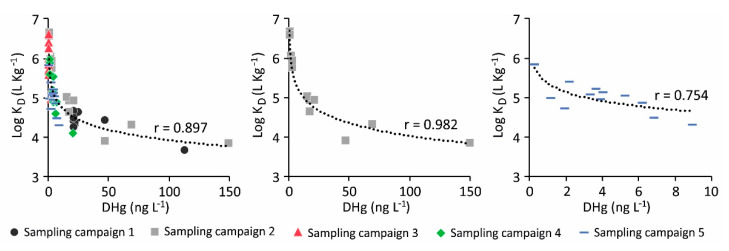
Relationship between logKD and dissolved DHg (ng L^−1^) in surface and bottom water samples collected at the six investigated sites in the Bay of Panzano during sampling campaigns 1–5 performed under different environmental conditions. Best fit was obtained using a 2nd-order polynomial curve.

**Table 1 ijerph-18-10192-t001:** Environmental conditions and Isonzo River daily discharge (m^3^ s^−1^) during the six sampling campaigns performed to collect surface and bottom water samples at the six investigated sites in the Bay of Panzano.

Sampling Campaign		Environmental Conditions	Isonzo River Daily Discharge (m^3^ s^−1^)	Sampling Sites
1	1 February 2016	unperturbed	4.45	P1–P6
2	2 March 2016	Isonzo River high discharge	328	P1–P6
3	31 March 2016	windy sea	45.8	P1–P6
4	21 April 2016	unperturbed	39.0	P1–P6
5	4 May 2016	Isonzo River moderate discharge	87.3	P1–P6
6	15 April 2016	after a ship with large draught	72.0	A–E and P3

**Table 2 ijerph-18-10192-t002:** Ranges of Hg concentration in surface sediments from this study compared to local areas of the Gulf of Trieste and other similar environments in the world as reported in the literature.

Location	Hg (µg g^−1^)
Bay of Panzano (Italy)–this study	0.77–6.39
Bay of Panzano (Italy)–[30]	0.30–13.5
Bay of Panzano (Italy)–stations D0, D2, E3–[52]	1.40–5.54
Timavo estuary (Italy)–[53]	0.08–2.40
Isonzo River mouth (Italy)–[23]	2.76–10.2
Isonzo River mouth (Italy)–stations A4, IS1, IS2–[52]	4.92–6.18
Marano and Grado Lagoon (Italy)–[17] *	0.68–9.95
Grado Lagoon (Italy)–[54] **	9.50–14.4
Marano Lagoon (Italy)–[17]	1.22–4.49
Aussa River, Marano Lagoon (Italy)–[55]	0.82–5.69
Gulf of Trieste (Italy)–stations A0, A2, A3–[52]	0.54–1.24
Gulf of Trieste (Italy)–[19]	0.10–23.3
Venice Lagoon (Italy)–[56]	0.50–2.51
Gulf of Trieste (Slovenia)–[57]	0.06–0.88
northern Adriatic Sea–[21]	0.05–8.63
central Adriatic Sea–[21]	0.02–0.13
southern Adriatic Sea–[21]	0.03–0.07
Northern Tyrrhenian Sea coastal area (NE Latium, Italy)–[58]	0.03–2.20
Thau Lagoon (France)–[59]	0.30–0.46
Port of Cartagena (Spain)–[60]	21.6–136
Port of Barcelona (Spain)–[60]	0.94–4.12
Port of Coruña (Spain)–[60]	0.54–6.41
Bay of Cádiz (Spain)–[61]	0.77–1.18
Gulf of Cádiz (Spain)–[61]	0.24–0.30
Nalón estuary (Spain)–[62]	0.10–1.33
Jiaozhou Bay (China)–[11]	0.02–0.15
Jinzhou Bay (China)–[11]	0.80–25.0
Chesapeake Bay (USA)–[63]	0.08–0.18

* 0–1 cm surface sediment; ** 0–10 cm surface sediment.

**Table 3 ijerph-18-10192-t003:** Ranges of Hg concentration in the dissolved fraction (DHg, ng L^−1^) and in the SPM (PHg, µg g^−1^ and ng L^−1^) from this study compared to local areas of the Gulf of Trieste and other similar environments as reported in the literature.

Location	Water Layer	DHg (ng L^−1^)	PHg (ng L^−1^)	PHg (µg g^−1^)
Bay of Panzano (Italy)–this study	surface	<LOD–46.8	0.39–12.5	0.10–1.55
	bottom	<LOD–149	2.42–3.73	0.11–1.87
Isonzo River mouth (Italy)–[22]	surface	0.46–17.0	<LOD–97.6	0.09–37.3
	bottom	1.38–81.3	<LOD–92.7	n.s.
Isonzo River mouth (Italy)–[23]	surface	n.d.	n.s.–112	1.20–2.70
	bottom	n.d.	<1.00–84.0	0.20–2.40
Isonzo River mouth (Italy)–[36]	surface	6.25	105	12.1
	bottom	<LOD–4.16	10.2–57.3	1.45–1.59
Timavo estuary (Italy)–[53]	surface	<LOD–2.27	2.91–4.30	1.96–4.28
	bottom	<LOD–5.10	8.80–23.1	1.78–2.57
Gulf of Trieste (Italy)–[69]	surface	<LOD–4.90	n.d.	0.05–0.56
	bottom	0.18–2.69	n.d.	
Grado Lagoon inlet (Italy)–[70]	water column	5.30–8.10	5.80–160	n.s.
Aussa River, Marano Lagoon (Italy)–[55]	surface	4.10–52.4	3.40–131	1.30–20.3
	bottom	9.10–48.0	4.80–32.1	1.10–2.20
Tagus Estuary (Portugal)–[71]	water column	3.61–65.4	n.d.	0.36–8.63
Portuguese coast (Portugal)–[72]	surface	1.00–28.1	n.d.	0.80–15.3
Tinto estuary (Spain)–[61]	surface	1.66–19.8	n.d.	0.06–1.12
Odiel estuary (Spain)–[61]	surface	3.29–40.0	n.d.	0.14–6.52
Gulf of Cádiz (Spain)– [61]	surface	0.42–0.76	n.d.	0.08–0.42
Nalón estuary (Spain)–[73]	surface	2.13-5.09	2.59–9.10	0.25–2.35
	bottom	0.74–7.73	3.05–14.4	0.43–2.69
Jiaozhou Bay (China)–[11]	surface	2.47–9.87	n.d.	0.06–0.47
New York/New Jersey Harbor–[7]	surface	0.34–1.00	2.91–64.6	0.08–1.78
New York/New Jersey Harbor–[7]	bottom	<LOD–3.01	4.47–295	0.19–2.43

n.d.: not determined; n.s.: not specified.

**Table 4 ijerph-18-10192-t004:** LogK_D_ (L kg^−1^) for Hg in the surface and bottom water samples collected at the six investigated sites in the Bay of Panzano under different environmental conditions.

Sampling Campaign	Water Layer	P1	P2	P3	P4	P5	P6
1—1 February	surface	4.45	4.41	4.64	4.65	4.34	4.41
	bottom	4.70	4.69	4.38	4.52	3.69	4.29
2—2 March	surface	3.92	4.64	4.96	6.07	5.02	5.77
	bottom	4.33	3.88	4.96	5.93	6.68	6.58
3—31 March	surface	4.97	5.74	5.99	6.42	6.01	5.86
	bottom	6.22	5.68	6.61	6.28	5.70	5.60
4—21 April	surface	4.88	5.14	4.60	5.52	4.88	5.16
	bottom	5.10	5.60	5.18	5.88	4.11	5.96
5—4 May	surface	5.12	5.06	5.83	4.48	4.72	4.32
	bottom	4.86	5.03	5.22	4.97	4.93	5.40

## Data Availability

The data presented in this study are available on request from the corresponding author.

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
