# Peer review of "The Legacy of the Idrija Mine Twenty-Five Years after Closing: Is Mercury in the Water Column of the Gulf of Trieste Still an Environmental Issue?"

_ijerph, 2021, doi:10.3390/ijerph181910192_

Round 1

Reviewer 1 Report

Authors analyzed and discussed the mercury pollution from four aspects, determined that the source of Hg pollution in the Gulf of Trieste was Isonzo/Soča River freshwater inputs. The overall research idea is clear, but there are several small problems. The comments are as follows:

  1. Page2, line 43 The Hg content in the Mediterranean can indicate that Hg is enriched globally, Why? Please explain.
  2. Page2, line 44 There is an incomplete parenthesis. The same question in Page3, line 48.
  3. Please add a literal description of the location relationship between Idrija mine and the study area, with a view to comparing the Figure 1.
  4. Page5, Table 1Is the Isonzo River discharge mean value? If so, please specify in the paper.
  5. Page5, Section 3.1Please supplement the significance of different physico-chemical parameters for the study of mercury pollution.
  6. Please add the reason for the monthly difference in turbidity, and for the differences in sampling points of Grain-sizeof surface sediments(eg. Outliers P5).
  7. Page5, Table 2 Please explain the reasons for the large difference ofHg content in the first three studies.
  8. Conclusions part need simplify.

Reviewer 2 Report

Dear author:

Do the samples are representative? Please, indicate the statistical analysis of the data.

Table 4. Indicate the value of the LogKD with standard deviation.

How many samples were taken at each site? For example, in figure 3A) the standard deviation is not indicated.

Indicate the number of samples considered for the grain size composition figure.

Does the data from this study are conclusive and reproducible?

Author Response

Dear author:
Do the samples are representative? Please, indicate the statistical analysis of the data.
As also suggested by Reviewer #3, multivariate analysis was performed. In detail, principal component
analysis (PCA) was employed for the evaluation of the relationships within samples, within variables and
between samples and variables.
We have added the section 2.5 Exploratory Multivariate Data Analysis in the 2. Materials and Methods
section, and the Figure S3 showing the output summarising the results of PCA performed on physico-chemical
parameters, dissolved and particulate Hg observed in surface and bottom water samples collected at the six
investigated sites (P1-P6) during sampling campaigns 1-5 performed under different environmental
conditions.
We have modified the sections 3.3 Suspended Particulate Matter: Distribution and Hg Concentrations and
3.4 Dissolved Hg by adding the references to the PCA output (Fig. 4 in the revised version of the manuscript)
which confirmed what has been already stated in the text.
In detail, the PCA output confirmed that:
- The SPM distribution in the investigated area depends heavily on the river discharge and that the
highest concentrations of PHg were observed under perturbed conditions during sampling campaign
2 (perturbed conditions, high river discharge). 

- The amount of PHg observed during sampling campaign 5 (perturbed conditions, moderate river
discharge) was comparable to that of the sampling campaigns performed under conditions of windy
sea (sampling campaign 3) and under unperturbed conditions (4).
- Relatively elevated values of PHg in the investigated area may be restricted to brief periods of
particularly intense discharge from the Isonzo River
- The highest concentrations of DHg were detected in winter both under unperturbed (sampling
campaign 1)

Table 4. Indicate the value of the LogKD with standard deviation.
As already reported in the text, Hg distribution coefficients (KD, L kg−1) were calculated as the ratio between
Hg concentration in the SPM (PHg, µg g−1) and in the dissolved fraction (DHg, ng L−1) and expressed on a
logarithmic scale. However, we cannot indicate the value of the logKD with standard deviation since two
water samples were collected at each site during every sampling campaign: one from the surface water layer
and one at the bottom of the water column.
We have specified in the text (section 2.2 Sampling strategy, lines 128-130 in the revised version of the
manuscript) that two water samples were collected from surface (0-0.5 m depth) and bottom (0.50 cm from
the bottom) water layers, respectively.

How many samples were taken at each site? For example, in figure 3A) the standard deviation is not
indicated.
Indicate the number of samples considered for the grain size composition figure.
Figure 3A is about Hg concentration in the surface sediments. Standard deviation is not indicated since we
collected one sample at each site. As already reported in the section 2.2 Sampling strategy, surface sediments
were collected at each site during sampling campaign 1. In detail, surface sediments were collected by means
of a stainless steel grab and three distinct aliquots of sediment were collected and homogenised in situ to
get a composite sample. We added this detail in the text (section 2.2 Sampling strategy, lines 148-152 in the
revised version of the manuscript).
Regarding DHg and PHg, please, see our response to your previous comment.

Does the data from this study are conclusive and reproducible?
We retain that results from this study are of great interest from a management point of view related to future
dredging operations to allow navigation of ships approaching the port area of Monfalcone.
This manuscript is focused on the legacy of one of the largest Hg mines worldwide (Idrija, located in the
Slovenian sector of the Isonzo River drainage basin) in terms of Hg distribution along the water column in the
northernmost sector of the Gulf of Trieste. Due to over 500 years of mining activity, the marine-coastal area
of the Gulf was found to be contaminated by Hg, entering the Gulf mainly in association with the suspended
particulate matter transported by the Isonzo River.
Although several studies have been focused on the sediments of the Gulf (e.g. Covelli et al., 1999, 2001;
Acquavita et al., 2010; Emili et al., 2014, 2016), there was a lack of knowledge concerning the behaviour of
Hg along the water column. This study fills this gap providing a baseline for both dissolved and particulate Hg
along the water column and demonstrating that the occurrence of Hg in the investigated area remains an
issue of environmental concern especially following periods of high discharge from the Isonzo River, which
was confirmed as the main source of Hg in the Gulf.
We also retain that all the necessary information concerning the sampling and laboratory (sample
preparation and analysis) activities were appropriately provided in order to make other researchers able to
understand and reproduce how the study was carried on.

Reviewer 3 Report

Good and clearly presented study. review of methods shows good analysis and quality control. interesting study. Some suggestions to make the work a bigger impact.

Tables 2 and 3 might be better presented as figures showing the range and grouped by country??

would there be useful assessment if there was a bit of multivariate analysis? other parameters: pH, salinity, organic matter, temperature, flow, turbidity may contribute differently. All we have been presented with is a single relationship with flow.

any archive e.g. sediment core data to show historical behaviour of Hg supply in the region? would be nice to try and put your time series in context?

Reviewer 4 Report

Please see the notes in the pdf file and in the file "reviewer comments to author manuscript".
